# Neurophysiological Hallmarks of Axonal Degeneration in CIDP Patients: A Pilot Analysis

**DOI:** 10.3390/brainsci12111510

**Published:** 2022-11-07

**Authors:** Dario Ricciardi, Federica Amitrano, Armando Coccia, Vincenzo Todisco, Francesca Trojsi, Gioacchino Tedeschi, Giovanni Cirillo

**Affiliations:** 1I Division of Neurology and Neurophysiopathology, Department of Medical and Surgical Sciences, University of Campania “Luigi Vanvitelli”, Via L. Armanni 5, 80138 Naples, Italy; 2Department of Information Technologies and Electrical Engineering, University of Naples “Federico II”, 80125 Naples, Italy; 3Laboratory of Morphology of Neuronal Networks & Systems Biology, Division of Human Anatomy, Department of Mental and Physical Health and Preventive Medicine, University of Campania “Luigi Vanvitelli”, Via L. Armanni 5, 80138 Naples, Italy

**Keywords:** chronic inflammatory demyelinating polyneuropathy, EMG, axonal degeneration, motor unit analysis, subcutaneous immunoglobulin

## Abstract

In this work, we aim to identify sensitive neurophysiological biomarkers of axonal degeneration in CIDP patients. A total of 16 CIDP patients, fulfilling the clinical and neurophysiological criteria for typical CIDP, treated with subcutaneous immunoglobulin (ScIg) (0.4 g/kg/week) were evaluated at baseline (before ScIg treatment) and after long-term treatment with ScIg (24 months) by clinical assessment scales, nerve conduction studies (NCS) and electromyography (EMG). Conventional and non-conventional neurophysiological parameters: motor unit potential (MUP) analysis, MUP thickness and size index (SI)] and interference pattern (IP) features were evaluated after long-term treatment (24 months) and compared with a population of 16 healthy controls (HC). An increase of distal motor latency (DML) and reduced compound motor action potential (CMAP) amplitude and area in CIDP patients suggest axonal damage of motor fibers, together with a significant increase of MUP amplitude, duration and area. Analysis of non-conventional MUP parameters shows no difference for MUP thickness; however, in CIDP patients, SI is increased and IP area and amplitude values are lower than HC. Despite clinical and neurophysiological improvement after ScIg treatment, neurophysiological analysis revealed axonal degeneration of motor fibers and motor unit remodeling. Correlation analysis shows that the axonal degeneration process is related to the diagnostic and therapeutic delay. MUP area and SI parameters can detect early signs of axonal degeneration, and their introduction in clinical practice may help to identify patients with the worst outcome.

## 1. Introduction

Experimental, clinical and neurophysiological evidence has increasingly shown that chronic inflammatory demyelinating polyneuropathy (CIDP) represents a spectrum of disorders that are suspected to have diverse pathophysiological mechanisms, partly including dysimmunity, and showing different clinical phenotypes, variable disease course and response to specific treatments [1,2].

Despite this variability, clinical and electrophysiological recognition of the disease phenotypes [3,4] is critical for planning appropriate therapeutic management [5]. Recently, the detection of antibodies directed against structures of the nodal and paranodal regions of the nerve fibers has revealed new possible pathophysiological mechanisms [6]. Disease duration, the persistence of the immune attack and demyelination of the peripheral nerve fibers produce first axonal dysfunction and then secondary axonal degeneration [7,8].

Axonal degeneration represents one of the main causes of muscular atrophy and disability, with great impact on the patients’ quality of life [9]. Therefore, prevention of axonal loss and muscular wasting represents the most important therapeutic objective, influencing the long-term functional prognosis of CIDP patients. Lack of a reliable diagnostic biomarker means the diagnosis of CIDP is based on combined clinical and ancillary test results, including serological, cerebrospinal fluid (CSF), electrophysiological and neuroimaging data [10]; on the other hand, lack of accurate prognostic tools, together with clinical heterogeneity, renders treatment demands challenging and warrants long-term monitoring with validated outcome measures.

In clinical practice, several clinical and neurophysiological parameters are commonly used to predict the therapeutic response and outcome of CIDP patients. Validated clinical scales [11] to assess strength deficits (MRC and grip strength), disability (INCAT and I-RODS) and quality of life (EQ-5D) together with neurophysiological study represent, to the best of our knowledge, the cornerstone of the long-term monitoring of CIDP patients. Clinical observations have suggested that treatment of CIDP patients induces a recovery of clinical deficits and physiological nerve conduction, while others, being responders to the therapy, may indefinitely present abnormal nerve conduction parameters. Therefore, we strongly believe in the role and usefulness of nerve conduction study (NCS) during the follow-up to assess and substantiate the response to the treatment.

Recently, our group demonstrated that the neurophysiological monitoring of the amplitude of the distal compound motor action potential (dCMAP) and sensory nerve action potential (SNAP) correlates with the functional outcome in CIDP and with strength, disability [12] and quality of life scores [13]. Degeneration of motor axons prompted the analysis of the recruitment features and the study of motor unit architecture with both conventional and non-conventional parameters, giving an important contribution to the management of CIDP patients [14].

Therefore, in the present study, we aim to correlate clinical, conventional and non-conventional neurophysiological parameters to identify sensitive biomarkers of axonal degeneration.

## 2. Materials and Methods

### 2.1. Study Population and Treatment

This retrospective cohort study included 16 patients (10 male and 6 female) (mean age 61.13 ± 3.25) affected by typical CIDP, according to the EAN/PNS criteria [5], and responders to the first cycle of intravenous immunoglobulin (IVIg) (0.4 g/kg/day). Patients were then switched to subcutaneous immunoglobulin (ScIg) (0.4 g/kg/week) and treated at a stable dose for at least 24 months at the I Division of Neurology and Neurophysiopathology of the University of Campania “Luigi Vanvitelli” (between 2017 and 2021). The ScIg treatment for CIDP patients is included in the standard care protocol. During the whole study period, no concurrent treatment with steroids, plasmapheresis or other immunosuppressive medications was used. We excluded patients with atypical CIDP variants, history of alcohol abuse and chemotherapy and affected by diabetes mellitus, chronic kidney disease, carential (vit. B12) and other dysimmune disorders (lupus, rheumatoid arthritis, coeliac disease).

Neurophysiological parameters were also compared with a population of 16 age- and sex-matched healthy controls (HCs) (8 M and 8 F) (mean age 59.7 ± 5.33). HCs were referred to our division with suspicion of a neuromuscular disease not confirmed by proper clinical and neurophysiological examination.

All participants provided written informed consent with the protocol approved by the local ethics committee.

### 2.2. Clinical Parameters

CIDP patients were evaluated for clinical parameters using the following established assessment tools for patients with immune-mediated polyneuropathies at baseline (before IVIg and ScIg treatment) and after 24 months of ScIg treatment [15]. Clinical evaluation was performed independently and blinded (G.C.).

The Medical Research Council sum score (MRCS) was used to assess the overall strength. The score is a summation of the MRC grades (range 0–5) of the following muscle pairs: deltoid, biceps and wrist extensors (for the upper limbs), iliopsoas, quadriceps and tibialis anterior (for the lower limbs) [16]. The MRC sum score, therefore, ranges from 0 (complete paralysis) to 60 (normal strength).

The inflammatory neuropathy cause and treatment (INCAT) sensory sum score was used to assess the overall sensory functions (pinprick, vibration and two-point discrimination) in patients with immune-mediated polyneuropathies [17]. This scale evaluates different sensory modalities in the upper and lower limbs and ranges from 0 (normal sensation) to 20 (severe sensory deficit). Finally, the overall disability sum score (ODSS) was used to monitor the overall disability, providing a total score ranging from 0 (no disability) to 12 (severe disability) [17].

The Rasch-built overall disability scale (R-ODS) for immune-mediated neuropathies [18] and the overall disability sum scale (ODDS) [17] were used to estimate the degree of patients’ disability and the impact of the disease on daily life. R-ODS is a 24-item scale for daily activity and social participation limitations, ranging from 0 (not possible to perform) to 48 (possible without any difficulty), and was used to assess disability. ODDS focuses on the upper (from 0 to 5) and lower limb (from 0 to 7) functions: a score of 0 indicates no limitations, whilst a score of 5 or 7 indicates no purposeful movement.

Finally, quality of life (QoL) was evaluated with the EuroQol visual analogue scale (EQ-VAS), a quantitative health measure to record patient’s perception of health on a scale ranging from “best health” (score = 100) to “worst health” (score = 0) [19].

### 2.3. Nerve Conduction Studies and Motor Unit Potential Parameters

Neurophysiological parameters of nerve conduction (NC) were assessed at baseline and after 24 months of ScIg treatment, while motor unit conventional and non-conventional parameters were assessed after 24 months of ScIg treatment, using a synergy electromyography machine (Synopo, Milan, Italy), according to the guidelines of the American Association of Neuromuscular and Electrodiagnostic Medicine [20].

Skin temperature was maintained to at least 33 °C at the palm and 30 °C at the external malleolus [5]**,** and surface adhesive pre-gelled disk electrodes were used for recording the CMAP, using the belly-tendon montage. For motor nerve conduction, we recorded from two proximal (right musculocutaneous and left femoral) and two distal motor nerves (right deep fibular and left ulnar) of the upper and lower limbs the following neurophysiological parameters: distal motor latency (DML), CMAP amplitude (measured peak to peak) and area (area between first takeoff and return to baseline), motor conduction velocity (MCV) and F-wave latency (the two latter only for distal muscles). Electromyography (EMG) was carried out to collect and analyze MUPs using a concentric needle-electrode (CNE) (0.46 mm = 26 G, registration area 0.07 mm^2^) into the following four muscles: brachial biceps (BB), 1st dorsal interosseous (FDI), vastus lateralis (VL) and extensor digitorum brevis (EDB). We first evaluated the spontaneous activity (fibrillation potentials and/or positive sharp waves) and then recorded a total of 10 MUPs (each “captured” with the amplitude delay line to time-lock the potential and with the averaging technique) at minimal effort from three insertion sites for each muscle. To reduce the chance of recording MUPs belonging to the same motor unit, the needle was inserted perpendicularly into the muscle, recording from the depth to the superficial layers. The examiner carefully moved the needle until each MUP became sharp and loud, with the highest amplitude and the shorter rise time. For each muscle, we collected both conventional and non-conventual MUP parameters. Conventional parameters included: MUP duration, amplitude, area and phases. Due to inter-operator differences in signal selections, technical aspects (i.e., the distance of the needle), signal-to-noise ratio and patient collaboration, we included in the study the analysis of two non-conventional parameters, MUP thickness and size index (SI). The former, as the ratio between the MUP area and amplitude, is much less affected by needle position and distance and quantifies the “thickness” on visual assessment [21]. The latter is obtained by mathematically normalizing the MUP ratio to its amplitude and has been proved to better discriminate between diseased and healthy muscles [21,22]. We aim to identify subtle changes in the MU architecture which would otherwise be missed, thus improving the quality of electromyographic parameters analysis in CIDP patients.

### 2.4. Interference Pattern Analysis (IPA)

The overlap of MUPs, which represents the activity of the recruited motor units (MUs), leads to a signal referred to as the interference pattern (IP). Different methods of IPA can be used [23,24]**,** and several parameters, such as the size and the number of recruited MUs, the firing rates and duration, influence the shape of the IP [25]. Thus, the analysis of IP provides meaningful information about muscle clinical status to be used for both diagnostic and monitoring purposes [26].

IP was recorded using CNE in three different sites of needle position at the maximal voluntary contraction. The examiner (D.R.) selected the sharpest pattern with the greatest MUP amplitude and higher discharge frequency. Pain may limit the maximum force that the patient may exert, and this could reduce the area and the root mean square values. To overcome this possible limit, for each of the four muscles, HCs and CIDP patients were instructed to contract as hard and as fast as possible, performing at least five maximum contractions with five minutes of rest between each trial.

Various amplitude parameters may be extracted from the IP. Among these, the IP area and the root-mean-square (RMS) amplitude have been studied. IP area is one of the main parameters used in IP analysis and reflects the number and the size of motor units recruited at different force levels. The IP area is computed as the sum of all areas under each rectified phase of the signal. As IP is a continuous function of time, the IP area can be defined for a particular time interval ranging from $T_{1}$ to $T_{2}$ by the following formula:$$IP_{Area}=\overset{T_{2}}{\underset{T_{1}}{\int }}\left|y\left(t\right)\right|dt$$
where y(t) is the IP function of time and $[T_{1};T_{2}$] is the time window on which the area is computed. IP rectification is obtained by means of absolute value, while the sum of phase areas is estimated through integration.

In mathematics and its applications, the RMS is defined as the square root of the mean value of the squared values of the quantity taken over an interval. The RMS value of any function y=f(t) over the range $[T_{1};T_{2}$] can be defined as:$$RMS=\sqrt{\frac{1}{T_{2}-T_{1}}\int_{T_{1}}^{T_{2}}y^{2}\left(t\right)dt}$$

In electronics, the RMS value is the effective value of a varying voltage or current. It is the equivalent steady DC (constant) value that gives the same effect. The RMS value is, therefore, a parameter frequently chosen in the analysis of IP signal because it reflects the level of the physiological activities in the motor unit during contraction.

A custom-made Matlab software for the automatic analysis of the interference pattern has been developed. IP recordings exported to a text file (.txt) from the synergy electromyography machine are automatically imported in a Matlab environment by the software. IP signals are plotted and shown to allow an expert technician to interactively choose the time window of interest. The visual analysis ensures the selection of the IP window in which there is the maximum muscle contraction, characterized by both the highest amplitude and firing rate. Moreover, this method avoids signal artifacts that may not be automatically recognized by the software [27]. The selected window is divided into consecutive 300 ms intervals, with an overlap of 150 ms (Figure 1).

The software computes the IP area of each interval. The highest value found among them and the RMS of the corresponding IP interval are then saved in an Excel file for the following statistical analysis. Specifically, using this method, the IP area and the RMS of the 300 ms interval corresponding to the highest muscle contraction are assessed starting from the recorded IP signal [26]. The area under the curve is computed by means of Matlab function ‘trapz’. It performs numerical integration via the trapezoidal method, which provides the approximation of the integration over an interval by breaking the area down into trapezoids with more easily computable areas that are finally summed up. We analyzed these two different features of the IP in both CIDP patients and HCs.

### 2.5. Statistical Analysis

Clinical data (MRCS, INCAT, R-ODS, ODSS and EQ-VAS scores) and neurographic parameters (DML, distal CMAP amplitude and area) were evaluated at baseline and after 24 months. EMG conventional (MUP duration, amplitude, area and phases) and non–conventional (MUP thickness and size index) parameters were assessed at 24 months of ScIg treatment. Categorical variables were expressed as a number, whilst continuous variables were expressed either as mean and standard deviation (SD) or median and interquartile range (IQR) according to their distribution appropriately assessed by the Shapiro–Wilk test. Between groups, differences were assessed for categorical variables, either by the Fisher Exact Test or the Chi-square test, with Yates’s correction, as appropriate. As for continuous variables, either a student T-test or a non-parametric Mann–Whitney U test were performed, according to their distribution.

Regression analysis (Pearson’s linear correlation) was applied for the correlation of neurophysiological parameters with some demographic variables, such as the time elapsed between the onset of the disease and the start of treatment.

Data were analyzed using Sigma Plot 10.0 software and expressed as mean ± standard deviation (SD), with *p* < 0.05 considered significant, using the Bonferroni method for multiple comparisons.

## 3. Results

### 3.1. Clinical Evaluation

Table 1 shows the demographic data of CIDP patients and HCs, while clinical data at baseline and after 24 months of ScIg treatment are presented in Table 2.

ScIg treatment is effective on strength recovery, as shown by the significant increase of MRC sum score after ScIg treatment for 24 months (** *p* ≤ 0.001). Changes in INCAT, R-ODS, ODSS and EQ-VAS scores after ScIg treatment suggest a recovery of somatosensory functions (INCAT), a reduction of the global disability (R-ODS, ODSS) and improvement of the perceived quality of life (QoL) (EQ-VAS), with reduction of the impact of the disease on the performance of daily life activities (** *p* ≤ 0.001).

All together, these data indicate that ScIg treatment is effective and well tolerated in CIDP patients.

### 3.2. Analysis of Nerve Conduction Parameters and Electromyographic Activity

Analysis of the nerve conduction parameters, including area and amplitude of CMAP and DML, in HCs and CIDP patients is shown in Table 3 and Figure 2.

Despite the progressive improvement of the MCV and F-wave after treatment, DML is increased and CMAP amplitude and area are lower in CIDP patients compared to HCs (* *p* ≤ 0.05; ** *p* ≤ 0.001). These neurographic features are indicative of axonal damage of motor fibers.

To further analyze the changes of the motor unit, we analyzed the electrophysiologic features of MUPs (phases, amplitude, duration and area) during electromyographic recordings (Table 4 and Figure 3).

Pathological spontaneous activity was not recorded in any of the four muscles both in CIDP patients and HCs. No significant difference was detected for MUP phases; in contrast, our analysis showed a significant difference of MUP amplitude, duration and area in all the target muscles of ScIg-treated patients compared to HCs (* *p* ≤ 0.05; ** *p* ≤ 0.001). These data indicate a chronic remodeling of MU after axonal damage and is supportive of neurogenic damage.

Analysis of non-conventional MUP parameters is presented in Table 4 and Figure 3. MUP thickness showed no difference between groups; however, size index is increased in CIDP patients, with a significant difference between groups (** *p* ≤ 0.001) (Figure 3A). Analysis of interference pattern (IP) data in all the target muscles revealed lower and significant IP area and IP RMS amplitude values in CIDP patients compared to HCs (* *p* ≤ 0.05; ** *p* ≤ 0.001) (Table 4, Figure 3B,C).

All together, these data suggest that despite the progressive clinical improvement, CIDP patients show neurographic marks of axonal degeneration of motor fibers and electromyographic signs of remodeling of the smaller MU and degeneration of the largest.

### 3.3. Clinical-Neurophysiological Correlation Analysis

Linear regression analysis was used to correlate neurophysiological parameters with clinical data. Diagnostic and therapeutic delay (i.e., disease duration) positively correlated both with CMAP area of Ul (*p* = 0.004; R = 0.77) and DP (*p* = 0.011; R = 0.72) nerves and with MUP area of all the examined muscles (*BB*, *p* < 0.05, R = 0.69; *FDI*, *p* = 0.0421, R = 0.62; *VL*, *p* = 0.001, R = 0.55; *EDB*, *p* = 0.0197, R = 0.74).

These data support a neurogenic remodeling of MU due to axonal damage in the distal nerve and muscles.

### 3.4. Logistic Regression Analysis

To identify the more sensitive MUP parameters in discriminating CIDP patients from HCs, we conducted logistic regression analysis. Our model includes the MUP conventional parameters (phases, amplitude, duration and area), is statistically significant (*p* < 0.0001) and capable of discriminating CIDP patients from HCs. MUP area represents the most discriminative (*p* = 0.05, for all muscles), as shown by ROC (receiver operating characteristic) curves (Figure 4). For non-conventional MUP parameters, regression analysis indicates the size index (Figure 4) as a good discriminator between CIDP patients and HCs (*p* < 0.0001).

## 4. Discussion

It is well established that long-term treatment with ScIg in CIDP patients leads to both clinical and neurophysiological improvement. Correlation of clinical parameters with neurophysiological features suggests that disability and QoL correlate with dCMAP and SNAP amplitudes, underlying their prognostic value and the importance of neurophysiological monitoring [12,13]. Long-term prognosis of treated CIDP patients depends on the therapeutic response and the occurrence of disability, mainly characterized by weakness and muscular atrophy [28]. Although CIDP is an acquired demyelinating disease, secondary axonal degeneration is commonly observed [8] and mainly related to therapeutic unresponsiveness [29,30]. Mechanisms of axonal damage are largely unknown; in the acute phase of the disease, demyelination and node/paranodal dysfunction may lead to Wallerian degeneration phenomena, while in the chronic phase the persistence of inflammatory and immune attack may lead to progressive degeneration of nerve fibers [31].

Demonstration of axonal loss of sensory and motor fibers was proved using sural nerve biopsy [32] and neurophysiological tolls, such as motor unit number estimation (MUNE) and motor unit number index (MUNIX), that were both decreased in CIDP patients [13,33,34]. Despite clinical and neurophysiological improvement, residual alterations of nerve conduction parameters after ScIg treatment for 24 months (increased DML and reduced CMAP amplitude and area), in the absence of demyelinating features (conduction blocks and temporal dispersion), suggest an axonal degeneration process [35].

Accordingly, our work demonstrated changes of conventional and non-conventional MUP parameters after treatment with ScIg, expression of motor axonal loss and remodeling of the MUPs. Changes of MUP amplitude, duration and area, in the absence of spontaneous activity and MUP phases increase, suggest stability of the MU remodeling after treatment. Analysis of two non-conventional parameters (MUP thickness and size index—SI) allowed an objective, reliable and accurate evaluation of the axonal loss. MUP thickness, as the ratio between MUP area and amplitude, is much less affected by technical bias (needle position) and SI. The normalization of the MUP ratio to its amplitude has been proved to better discriminate between diseased and healthy muscles [22]. Despite both being altered in CIDP patients, only the SI represents a sensitive parameter in detecting early signs of axonal degeneration. Axonal degeneration involving the largest motor unit of both proximal and distal muscles is also demonstrated by significantly lower mean firing rates at high contraction intensities, higher mean firing rates at low contraction intensities [36] and reduced IP area and IP RMS amplitude by interference pattern analysis (IPA).

Long-term severe disability of CIDP patients has also been correlated to the start delay of therapy [37], as also suggested by our clinical-neurophysiological correlation analysis showing that diagnostic and therapeutic delay is positively correlated both with CMAP amplitude and MUP area of all the examined muscles.

Despite significant results, our work has some limitations worth noting. First, we are aware that increasing the number of CIDP patients with more homogeneous clinical and neurophysiological features, shorter disease duration and treatment delay and longer clinical and electrophysiological follow-up would make our results and conclusions stronger. Second, we are aware that we have used uncommon and personalized neurophysiological analyses that need more studies to be verified. Third, we analyzed the motor unit and IP features after 24 months of ScIg treatment and not at baseline. We are not aware if those parameters were already abnormal at the time of the diagnosis or developed during the disease course. However, the lack of difference in the number of MUP phases between HCs and CIDP patients, as well as the absence of spontaneous activity, allows us to hypothesize a stability of the MU remodeling after ScIg treatment. Fourth, we included in the analysis the extensor digitorum brevis muscle that may present para-physiological abnormalities even in HCs; however, we needed to apply the same electrophysiological protocol for both upper and lower limbs. Finally, we are aware of the different clinical response and pathophysiological mechanisms (inflammatory, demyelinating or dysimmune) in CIDP variants and after other therapeutic strategies than IVIg or ScIg, such as corticosteroids, plasma exchange or rituximab [31], that might have a different impact on axonal degeneration and produce different results.

## 5. Conclusions

Despite these limitations, our analysis of conventional/non-conventional neurophysiological parameters highlights axonal degeneration of motor fibers in CIDP patients despite ScIg treatment. MUP area and SI parameters are useful in detecting early signs of axonal degeneration, and their introduction in clinical practice may help to identify patients with the worst outcome.

Preventing axonal degeneration, therefore, still represents one of the main targets of the CIDP therapy and may correlate with the best long-term clinical outcome.

## Figures and Tables

**Figure 1 brainsci-12-01510-f001:**
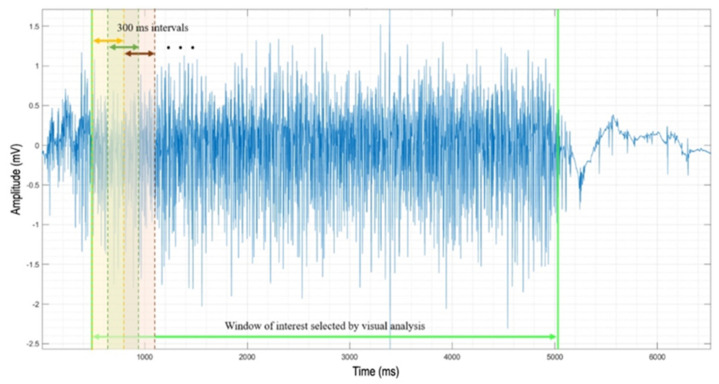
Exemplificative interference pattern analysis from BB muscle: the signal window between the two green vertical lines is selected by visual analysis by an expert technician. IP area is then computed on each 300 ms window. The window with the highest IP area is selected for the analysis.

**Figure 2 brainsci-12-01510-f002:**
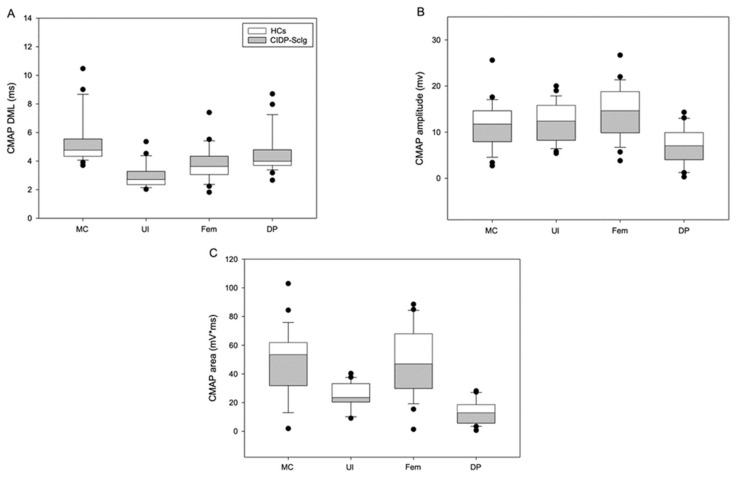
(**A**–**C**) Analysis of nerve conduction parameters (DML and CMAP amplitude and area) in CIDP patients after ScIg treatment and HCs (CIDP patients vs. HCs). Figure legend: HCs: healthy controls; CIDP: chronic inflammatory demyelinating polyneuropathy; ScIg: subcutaneous immunoglobulin; MC: right musculocutaneous; Ul: left ulnar; Fem: left femoral; DP: right deep fibular; DML: distal motor latency; CMAP: compound muscle action potential.

**Figure 3 brainsci-12-01510-f003:**
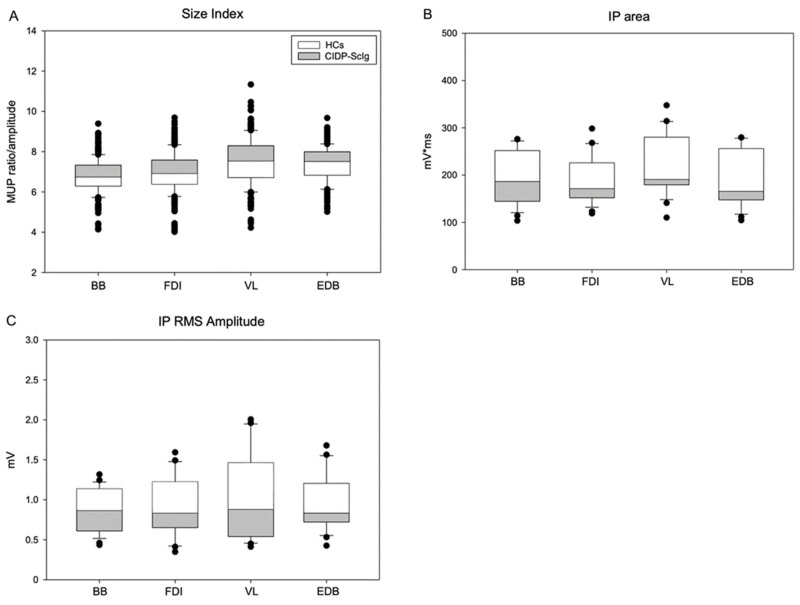
(**A**) Size index and (**B**,**C**) interference pattern analysis (IPA) in CIDP patients after ScIg treatment and HCs. Figure legend: HCs: healthy controls; CIDP: chronic inflammatory demyelinating poly-neuropathy; ScIg: subcutaneous immunoglobulin; BB: brachial biceps; FDI: first dorsal interosseous; VL: vastus lateralis; EDB: extensor digitorum brevis; MUP: motor unit potential; IP: interference pattern; RMS: root mean square.

**Figure 4 brainsci-12-01510-f004:**
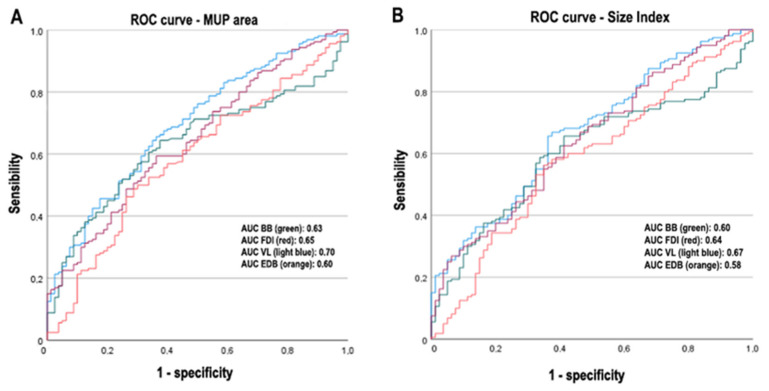
ROC curves for MUP area and size index (CIDP patients vs. HCs). Figure legend: BB: brachial biceps; FDI: first dorsal interosseous; VL: vastus lateralis; EDB: extensor digitorum brevis; MUP: motor unit potential; AUC: area under the curve.

**Table 1 brainsci-12-01510-t001:** Demographic data of CIDP patients and HCs.

Parameters	CIDP Patients	HCs
Age (years)	61.13 ± 3.25	59.7 ± 5.33
Gender (M/F)	10/6	8/8
Disease duration (months) at diagnosis	70.86 ± 12.96	/
Delay of ScIg treatment from disease onset (months)	27.06 ± 7	/
ScIg dose (g/kg/week)	0.2–0.4	/
Mean dose ScIg (g/week)	21.25 ± 2.23	/

Table legend. CIDP: chronic inflammatory demyelinating polyneuropathy; HCs: healthy controls; ScIg: subcutaneous immunoglobulin; EMG: electromyography.

**Table 2 brainsci-12-01510-t002:** Clinical parameters at baseline (before treatment), after one cycle of IVIg and after ScIg for 24 months.

Clinical Parameters/Cut-Off	Baseline	IVIg Treatment	ScIg Treatment	*p*
MRCS/60	31.3 ± 1.0	51.3 ± 0.9	56.5 ± 0.6	<0.01
INCAT/20	9.5 ± 0.7	6.7 ± 0.8	5.6 ± 0.9	<0.01
R-ODS/48	26 ± 1.9	35.2 ± 4.6	42.7 ± 3.1	<0.01
ODSS/12	7.2 ± 0.6	5.3 ± 0.9	3.0 ± 0.5	<0.01
EQ-VAS/100	34.2 ± 3.1	68.9 ± 2.8	79.4 ± 4.9	<0.01

Table legend. ScIg: subcutaneous immunoglobulin; MRCS: Medical Research Council sum score; INCAT: inflammatory neuropathy cause and treatment; R-ODS: Rasch-built overall disability scale; ODSS: overall disability sum scale; EQ-VAS: EuroQol visual analogue scale. Baseline vs. ScIg treatment.

**Table 3 brainsci-12-01510-t003:** Distal motor latency (DML) and CMAP amplitude and area of CIDP patients after ScIg treatment and HCs.

	CIDP Patients				HCs			
Nerve	MC	Ul	Fem	DP	MC	Ul	Fem	DP
DML (ms)	5.8 ± 0.4	3.2 ± 0.2	4.1 ± 0.3	4.8 ± 0.4	4.3 ± 0.01	2.4 ± 0.01	3.1 ± 0.1	3.7 ± 0.2
*p*	<0.05	<0.01	<0.05	<0.01				
CMAP Amplitude (mV)	9.7 ± 1.0	10.2 ± 0.8	12.2 ± 1.3	5.6 ± 0.9	15.0 ± 1.6	16.2 ± 0.8	18.6 ± 1.6	9.6 ± 1.1
*p*	<0.01	<0.01	<0.01	<0.01				
CMAP Area (mV*ms)	43.8 ± 5.2	21.2 ± 1.7	42.2 ± 4.7	11.2 ± 2.0	60.9 ± 7.6	32.5 ± 2.4	62.8 ± 9.1	18.8 ± 1.8
*p*	<0.01	<0.01	<0.05	<0.05				

Table legend. CIDP: chronic inflammatory demyelinating polyneuropathy; HCs: healthy controls; MC: right musculocutaneous; Ul: left ulnar; Fem: left femoral; DP: right deep fibular; DML: distal motor latency; CMAP: compound muscle action potential; CIDP patients vs. HCs.

**Table 4 brainsci-12-01510-t004:** “Conventional” and “non-conventional” MUP parameters in CIDP patients after ScIg treatment and HCs.

	CIDP Patients				HCs			
Muscle	BB	FDI	VL	EDB	BB	FDI	VL	EDB
Phases (n)	3.1 ± 0.2	3.4 ± 0.1	3.3 ± 0.1	3.7 ± 0.3	2.9 ± 0.1	3.3 ± 0.2	3.4 ± 0.1	3.3 ± 0.3
*p*	0.10	0.60	0.60	0.09				
Amplitude (mV)	0.8 ± 0.1	1.2 ± 0.1	1.4 ± 0.1	1.7 ± 0.3	0.5 ± 0.1	0.7 ± 0.1	0.9 ± 0.1	1.4 ± 0.2
*p*	<0.01	<0.01	<0.01	<0.01				
Duration (ms)	11.3 ± 0.2	10.7 ± 0.2	14.6 ± 0.3	11.4 ± 0.2	10.1 ± 0.2	9.6 ± 0.2	13.5 ± 0.2	10.2 ± 0.1
*p*	<0.05	<0.01	<0.05	<0.01				
MUP Area (μV*ms)	1160 ± 87	1759 ± 138	2629 ± 188	2474 ± 139	736 ± 42	1022 ± 73	1498 ± 94	2053 ± 163
*p*	<0.01	<0.01	<0.01	<0.05				
MUP Thickness	1.4 ± 0.05	1.4 ± 0.05	1.9 ± 0.06	1.5 ± 0.036	1.4 ± 0.06	1.3 ± 0.05	1.8 ± 0.07	1.4 ± 0.05
*p*	0.89	0.16	0.28	0.25				
Size Index	7.0 ± 0.06	7.4 ± 0.06	8.1 ± 0.06	7.8 ± 0.05	6.1 ± 0.05	6.0 ± 0.07	6.3 ± 0.07	6.5 ± 0.07
*p*	<0.01	<0.01	<0.01	<0.01				
IP Area (mV*ms)	169.5 ± 19.5	155.3 ± 15.5	178.9 ± 25.5	149.7 ± 22.8	254.6 ± 25.8	247.8 ± 23.0	294.6 ± 49.7	262.8 ± 27.8
*p*	<0.05	<0.01	<0.05	<0.01				
IP RMS (mV)	0.76 ± 0.1	0.71 ± 0.1	0.87 ± 0.1	0.74 ± 0.1	1.12 ± 0.1	1.33 ± 0.1	1.39 ± 0.2	1.43 ± 0.2
*p*	<0.05	<0.01	<0.05	<0.01				

Table legend. CIDP: chronic inflammatory demyelinating polyneuropathy; HCs: healthy controls; BB: brachial biceps; FDI: first dorsal interosseous; VL: vastus lateralis; EDB: extensor digitorum brevis; MUP: motor unit potential; IP: interference pattern; RMS: root mean square; CIDP patients vs. HCs.

## Data Availability

Not applicable.

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
