# Peer review of "Neurophysiological Hallmarks of Axonal Degeneration in CIDP Patients: A Pilot Analysis"

_brainsci, 2022, doi:10.3390/brainsci12111510_

Round 1

Reviewer 1 Report

General comments:   

The manuscript describes an exploratory effort in a smaller sample of CIDP patients.

There are several conceptual flaws and several important issues that warrant further clarification. 

Title: the title is misleading as the study design does not appear to be prospective, only few parameters were non-conventional and lack of alternate reference standard for axonal degeneration implies circular reasoning with regards to electrophysiologic findings.

Abstract:

The main study aim and design do not match, speculation on meaning study findings should be omitted, and conclusions are not supported by their data

Introduction

The authors state that CIDP is a heterogenous disorder, while in fact it is more accurate to note that it is a spectrum of disorders that are suspected to have diverse pathophysiological mechanisms involved that are partly dysimmune mechanisms. 

In contrast to authors statement, axonal degeneration is not the only process to cause muscle atrophy and disability in CIDP. Axonal dysfunction and demyelination are also known to be important causes of disability, and disuse may also result in muscle wasting. However, axon loss is known to be one of the important determinants for long term outcome [1]

The authors state that lack of sensitive biomarker impedes proper patient selection, but they appear to confuse 2 important aspects here: 

1.     A reliable diagnostic biomarker is lacking, consequently CIDP diagnosis is based on combined clinical and ancillary test results.  

2.     Accurate prognostic tools in CIDP are also lacking, combined with clinical heterogeneity this renders treatment demands challenging and warrants long-term monitoring with validated outcome measures. 

I would respectfully disagree with authors here that changes in electrophysiologic profiles are an accurate reflection of treatment response, and as authors speculate to be able to distinguish between patients that achieve remission and those that may need chronic active treatment.

The fact that dCMAP and SNAP correlate with strength, disability and quality of life scores does not imply that they can be used to predict individual disease course. 

The authors state to have aimed to study clinical and electrodiagnostic parameters to identify biomarkers of axonal degeneration. The major conceptual flaw here is the lack of an appropriate reference standard (e.g. histological) for axonal degeneration, electrodiagnostic testing is already an established tool to identify loss of motor and sensory axons, and fact that clinical parameters are by no means markers of axonal degeneration.

Methods:

The authors have only included a smaller sample of CIDP patients (n=16), which reflects the exploratory effort of the present study. 

The authors should omit pharmaceutical names of IVIg and ScIg in this section and consider including this entity in disclosures. The fact that patients were only included when they were treated at stable dose for 24 months after successful switch to ScIg, indicates selection of patients and not a prospective but rather retrospective cohort design. 

It is unclear which patients were eligible for inclusion (e.g. they state typical or did they also include variants of CIDP, which diagnostic classes ‘CIDP’ or also ‘possible CIDP’, previous treatments and disease duration, level of disability, etc.). This warrants more clear and detailed explanation for potential readers. 

The authors should also provide exclusion criteria used for the eligible patients and healthy controls. 

Clinical assessment

The authors have used several clinical scales (MRC, R-ODS, INCAT, QoL), but remains unclear if this was done by treating physicians or independent and blinded rater. 

NCS protocol

Did the authors sample uni-bilateral, if unilateral which side was selected?

Did the authors use standardized recording sites, including distances, and consider methods to reduce variability between baseline and follow-up measurements? If so, the authors should specify muscles and methods to variability between consecutive recordings within same patient. 

The authors state to heave used peak-peak CMAP amplitude assessment, but these are more susceptible to variation than the more common baseline-peak approach. 

How did the authors define CMAP area of the recorded responses (e.g. area between first take off and return to baseline or last return)?

Why did the authors select these nerves (ulnar, musculocutaneous, fibular and femoral)? Their combination is far from common in clinical setting of CIDP nor in published studies 

DML of musculocutaneous and femoral nerves are quite variable and their F-waves technically even more challenging, how did the authors solve these items? How did the authors even calculate MCV in femoral nerve? There are no accepted reference values for these measurements, nor are they common in routine clinical setting.  

Did the authors at needle myography select 10 MUPs from each of the 3 directions sampled in each muscle, or only 10 collectively from all 3 directions? 

Was there a single or multiple raters for NCS and needle myography, and where these blinded to clinical data and treatment status? 

How did the authors analyze the conventional MUP parameters, visually or quantitatively (including multi MUP analysis or turn/amplitude)? 

Did the authors perform needle EMG once, or also at follow-up. If the latter applied, how did they correct for differences in sample sites (actual site of insertion and sampling depth)? 

The authors have also used an uncommon approach of determining ratio between MUP area and amplitude, but in contrast to authors statement this is still dependent on recording distance [2]

Why did the authors use an unconventional interference pattern analysis, rather than the MUNIX and MUSIX to match their other assessments or even CMAP-scan?

The authors should provide more details on how they obtained their data for interference pattern analysis (e.g. did they used standard surface recording electrodes, including their sizes and specify standardized sites sampled, ect.). Moreover, they appear to have only selected maximal voluntary contraction, not other levels indicating that many superimposed signals were processed for their analysis.

In contrast to author’s statement, the surface recordings used for interference pattern analysis are all limited by several important technical aspects inherent to this technique and only allow an index to be obtained rather than accurate assessment of all involved MUP characteristics. 

Statistics

Did the authors actually consider p =0.05 significant, or is this a typo?

Results

In table 1 the authors mention a rather long disease duration (average >70 months) in their enrolled CIDP patients, long delay of ScIg initiation and longer treatment duration than the 24 months required for the study, abnormal ScIg dosing (mean 1633g/week).

In table 2, the authors mention rather low clinical scores and their p-values are misleading as the patients were already selected on objective improvement and stable course on fixed maintenance dosing. 

The electrodiagnostic results should be displayed in box plots not bars, as the latter lacks important information on distribution of electrodiagnostic parameters. 

Interpretation of data belongs to discussion section, not results, including speculation on neurogenic damage and MUP remodeling. 

The linear regression performed by the authors also suggests that disease duration and years without treatment were important determinants of CMAP area, confirming the potential selection bias of the included sample. 

The results of logistic regression are underwhelming and indicate little clinical value. 

Discussion:

This section lacks appropriate and balanced discussion, the first paragraph should summarize main study findings and their potential implications, successive paragraphs should discuss the study findings in more detail and compare these with other relevant studies previously published. Instead, the authors now have multiple paragraphs speculating on various aspects with no systematic comparison with previous work.

The study limitations warrant further expansion, and include more than small samples size, such as heterogeneity of patient characteristics, longer disease duration and treatment delay, selection of patients including those only stable on ScIg monotherapy, shorter follow-up (24 months in a disorder that is chronic and with patients having disease >70 months), uncommon NCS methodology (sites and parameters sampled, also see methods above), etc.

The conclusions do not match study aims, and indicate that treatment effect ScIg and delay of treatment initiation were main determinants here. 

1.         Al-Zuhairy, A., et al., Axonal loss at time of diagnosis as biomarker for long-term disability in CIDP.Muscle Nerve, 2022.

2.         Sonoo, M., et al., Updated size index valid for both neurogenic and myogenic changes. Muscle Nerve, 2020. 62(6): p. 735-741.

Author Response

Issues raised by Reviewer 4

Title: the title is misleading as the study design does not appear to be prospective, only few parameters were non-conventional and lack of alternate reference standard for axonal degeneration implies circular reasoning with regards to electrophysiologic findings.

  1. we thank the reviewer for these comments, we have rephrased the title, accordingly

Abstract:

The main study aim and design do not match, speculation on meaning study findings should be omitted, and conclusions are not supported by their data

R. we thank the reviewer for these comments, we have rephrased and revised the abstract

 Introduction

The authors state that CIDP is a heterogenous disorder, while in fact it is more accurate to note that it is a spectrum of disorders that are suspected to have diverse pathophysiological mechanisms involved that are partly dysimmune mechanisms. 

R. we thank the reviewer for this comment, we have rephrased the sentence

In contrast to authors statement, axonal degeneration is not the only process to cause muscle atrophy and disability in CIDP. Axonal dysfunction and demyelination are also known to be important causes of disability, and disuse may also result in muscle wasting. However, axon loss is known to be one of the important determinants for long term outcome [1]. 

R. we are aware of these aspects, we have rephrased the sentence.

The authors state that lack of sensitive biomarker impedes proper patient selection, but they appear to confuse 2 important aspects here: 

  1. A reliable diagnostic biomarker is lacking, consequently CIDP diagnosis is based on combined clinical and ancillary test results.  
  2. Accurate prognostic tools in CIDP are also lacking, combined with clinical heterogeneity this renders treatment demands challenging and warrants long-term monitoring with validated outcome measures.

R. we thank the reviewer for these important comments that we have fixed in the text, rephrasing the paragraph

I would respectfully disagree with authors here that changes in electrophysiologic profiles are an accurate reflection of treatment response, and as authors speculate to be able to distinguish between patients that achieve remission and those that may need chronic active treatment.The fact that dCMAP and SNAP correlate with strength, disability and quality of life scores does not imply that they can be used to predict individual disease course. 

R: the era of the precision medicine is still far from the actual medical practice, based on clinical evidence and tests’ results. In this context, positive changes of neurophysiological measures from a small group of patients that significantly improve their clinical condition after the proper therapy should be considered good indicators of therapeutic response and prognostic factors. Our experience suggests that positive correlation of dCMAP and SNAP amplitudes with clinical measures might indicate good prognosis. However, according to the reviewer’s comments, we have rephrased the paragraph.

The authors state to have aimed to study clinical and electrodiagnostic parameters to identify biomarkers of axonal degeneration. The major conceptual flaw here is the lack of an appropriate reference standard (e.g. histological) for axonal degeneration, electrodiagnostic testing is already an established tool to identify loss of motor and sensory axons, and fact that clinical parameters are by no means markers of axonal degeneration.

R: we are aware of the lack of histological findings that demonstrate axonal degeneration; however, electrodiagnostic testing represents an established technique to identify nerve fiber loss and clinico-electrophysiological-pathological correlations have been previously reported

 Methods:

The authors have only included a smaller sample of CIDP patients (n=16), which reflects the exploratory effort of the present study. 

R: we are aware of the small number of patients, as reported in a huge number of other published papers on a rare disease.

The authors should omit pharmaceutical names of IVIg and ScIg in this section and consider including this entity in disclosures.

R: we have omitted the pharmaceutical names.

The fact that patients were only included when they were treated at stable dose for 24 months after successful switch to ScIg, indicates selection of patients and not a prospective but rather retrospective cohort design. 

R: we thank the reviewer for this comment, we agree with this observation and have changed accordingly.

It is unclear which patients were eligible for inclusion (e.g. they state typical or did they also include variants of CIDP, which diagnostic classes ‘CIDP’ or also ‘possible CIDP’, previous treatments and disease duration, level of disability, etc.). This warrants more clear and detailed explanation for potential readers. 

R: we stated that the study included 16 patients affected by typical CIDP. Accordingly, to EAN/PNS 2021 CIDP dmiagnostic criteria, typical CIDP patients shows progressive or relapsing course, symmetric, proximal and distal muscle weakness of upper and lower limbs, and sensory involvement of at least two limbs, with symptoms and sings developing over at least 8 weeks and with absent or reduced tendon reflexes in all limbs. We excluded atypical variants (e.g. multifocal, distal, pure motor…).HCs were referred to our division with suspicion of a neuromuscular disease, not confirmed by proper clinical and neurophysiological examination.

The authors should also provide exclusion criteria used for the eligible patients and healthy controls. 

R: we have included the exclusion criteria

Clinical assessment

The authors have used several clinical scales (MRC, R-ODS, INCAT, QoL), but remains unclear if this was done by treating physicians or independent and blinded rater. 

R: clinical assessment was performed blinded by one of the authors (G.C.)

NCS protocol

Did the authors sample uni-bilateral, if unilateral which side was selected?

R: we choose unilateral examination (right deep peroneal – EDB, right musculocutaneos cutaneous – BB, left femoral – VL, left ulnare – ID). All patients and HCs did not referred or shown sings or symptoms of right/left radiculopathy, accordingly to the neurophysiological side approach.

Did the authors use standardized recording sites, including distances, and consider methods to reduce variability between baseline and follow-up measurements? If so, the authors should specify muscles and methods to variability between consecutive recordings within same patient. 

R: Neurophysiological study has been standardized for both patients and HCs. Nerve and muscles examination with the respective distance were standardized and respected for single patients and HCs both for single examination and or for follow up measurements.

The authors state to heave used peak-peak CMAP amplitude assessment, but these are more susceptible to variation than the more common baseline-peak approach. How did the authors define CMAP area of the recorded responses (e.g. area between first take off and return to baseline or last return)?

R: We choose peak-peak CMAP amplitude assessment according to normative data of our laboratory while, regarding Area only negative phase was analyzed, excluding repolarization phase more susceptible to variation. As stated, recording protocol was standardized fixing a constant distance between active and reference electrode.

Why did the authors select these nerves (ulnar, musculocutaneous, fibular and femoral)? Their combination is far from common in clinical setting of CIDP nor in published studies 

R: we have adopted a neurophysiological protocol including nerves that supply both proximal and distal muscles of the limbs, in order to correlate the measures with clinical data.

DML of musculocutaneous and femoral nerves are quite variable and their F-waves technically even more challenging, how did the authors solve these items? How did the authors even calculate MCV in femoral nerve? There are no accepted reference values for these measurements, nor are they common in routine clinical setting.  

R: we agree with the reviewer. While we have analyzed DML of musculocutaneous and femoral nerves, easy to detect, choosing gain at 1mV with the cut off at the onset of negative CMAP deflection, we did not analyze F-waves of CV for MC and femoral.  DML were matched between HCs and patients with a fixed distance between stimulation cathode and the active recording electrode (15 cm for femoral and 18 cm for MC).

Did the authors at needle myography select 10 MUPs from each of the 3 directions sampled in each muscle, or only 10 collectively from all 3 directions? 

R: we collect a total of 10 MUPs from three different directions inside the muscle

Was there a single or multiple raters for NCS and needle myography, and where these blinded to clinical data and treatment status? 

R: neurophysiological assessment was performed blinded by one of the authors (D.R.)

How did the authors analyze the conventional MUP parameters, visually or quantitatively (including multi MUP analysis or turn/amplitude)? 

R: quantitative software-based analysis

Did the authors perform needle EMG once, or also at follow-up. If the latter applied, how did they correct for differences in sample sites (actual site of insertion and sampling depth)? 

R. EMG was performed only at the follow-up and compared with the EMG of HCs.

The authors have also used an uncommon approach of determining ratio between MUP area and amplitude, but in contrast to authors statement this is still dependent on recording distance [2]. 

R: we are aware of this technical limit; we have stated that this techniques is much less affected by needle positions and distance.

Why did the authors use an unconventional interference pattern analysis, rather than the MUNIX and MUSIX to match their other assessments or even CMAP-scan?

R: we decided to use an unconventional IPA to find new measures from intramuscular recordings

The authors should provide more details on how they obtained their data for interference pattern analysis (e.g. did they used standard surface recording electrodes, including their sizes and specify standardized sites sampled, ect.). Moreover, they appear to have only selected maximal voluntary contraction, not other levels indicating that many superimposed signals were processed for their analysis.

In contrast to author’s statement, the surface recordings used for interference pattern analysis are all limited by several important technical aspects inherent to this technique and only allow an index to be obtained rather than accurate assessment of all involved MUP characteristics. 

R: we have included in the revised text the missing details.

Statistics

Did the authors actually consider p =0.05 significant, or is this a typo?

R: significant p < 0.05

Results

In table 1 the authors mention a rather long disease duration (average >70 months) in their enrolled CIDP patients, long delay of ScIg initiation and longer treatment duration than the 24 months required for the study, abnormal ScIg dosing (mean 1633g/week).

R: we’re sorry for the typos. We have specified that the disease duration is calculated at the time of the diagnosis, we have eliminated the ScIg treatment duration (patients were all analyzed after 24 months of treatment, but they all continued the treatment) and we corrected the value of the mean ScIg dose for week.

In table 2, the authors mention rather low clinical scores and their p-values are misleading as the patients were already selected on objective improvement and stable course on fixed maintenance dosing. 

R: we agree that the values are misleading. The baseline values refer to the patients’ clinical condition before IvIg therapy, this is the reason for the low scores. We have added the data after IvIg treatment, before starting the ScIg.

The electrodiagnostic results should be displayed in box plots not bars, as the latter lacks important information on distribution of electrodiagnostic parameters. 

R: we have changed the graphs in box plots.

Interpretation of data belongs to discussion section, not results, including speculation on neurogenic damage and MUP remodeling. 

R: thanks, we have rephrased

The linear regression performed by the authors also suggests that disease duration and years without treatment were important determinants of CMAP area, confirming the potential selection bias of the included sample. 

R: we agree with this comment, disease duration and years without treatment are major determinant of disease progression and nerve dysfunction and axonal degeneration.

Discussion:

This section lacks appropriate and balanced discussion, the first paragraph should summarize main study findings and their potential implications, successive paragraphs should discuss the study findings in more detail and compare these with other relevant studies previously published. Instead, the authors now have multiple paragraphs speculating on various aspects with no systematic comparison with previous work.

R: also according to other comments, we have completely rewritten and rephrased the discussion

The study limitations warrant further expansion, and include more than small samples size, such as heterogeneity of patient characteristics, longer disease duration and treatment delay, selection of patients including those only stable on ScIg monotherapy, shorter follow-up (24 months in a disorder that is chronic and with patients having disease >70 months), uncommon NCS methodology (sites and parameters sampled, also see methods above), etc.

The conclusions do not match study aims, and indicate that treatment effect ScIg and delay of treatment initiation were main determinants here. 

R: we thank the reviewer for these suggestions, we have rephrased accordingly.

  1. Al-Zuhairy, A., et al., Axonal loss at time of diagnosis as biomarker for long-term disability in CIDP.Muscle Nerve, 2022.
  2. Sonoo, M., et al., Updated size index valid for both neurogenic and myogenic changes.Muscle Nerve, 2020. 62(6): p. 735-741.

R: we have added the missing reference and discussed them in the revised draft

Reviewer 2 Report

Recent studies have demonstrated the efficacy of immunosuppressive/immunomodulatory treatment in CIDP patients with electrophysiological axonal lesions that do not meet electrophysiological criteria for demyelinating lesions. Furthermore, the evaluation of axonal lesions as biomarkers of axonal degeneration may be helpful in predicting functional outcome in this patient group.

Title: The title of the manuscript is properly.

1.     Introduction

The introduction is quite well written, providing a lot of well-known information. It could have included more detailed content. The reason why this study was necessary is clear from the introduction.

2.     Materials and Methods

The study group was not described carefully. What coexisting diseases were seen in the patients? How many patients with CIDP had diabetes mellitus, etc? How old were them, whta was the age distribution, more ptients are old or young? The age influences the periphreal nerve conditions, axon loss, and the results of neurophysiological test.

Electrophysiological methods were described in details.

3.     Results

The longitudinal assessment and the predictive value of the results of the baseline studies is well presented.

The predictive value of clinical parameters should be added.

All figures and tables are readable.

4.     Discussion

The discussion section should be shortened. The authors repeated the results without reference to the studies of other authors. They should discuss their points of view, and refer to the previous findings. We do not know whether the authors’ results are compatible or not with the previous studies. The discussion section must be rewritten.

The authors have given a reliable description of the limitations of this work.

5.     References

The references cited should be enlarged.

Author Response

Issues raised by Reviewer 1

The introduction is quite well written, providing a lot of well-known information. It could have included more detailed content.

R. we thank the reviewer for this comment, we have rephrased some sections of the introduction and added other details.

The study group was not described carefully. What coexisting diseases were seen in the patients? How many patients with CIDP had diabetes mellitus, etc? How old were them, what was the age distribution, more patients are old or young? The age influences the peripheral nerve conditions, axon loss, and the results of neurophysiological test.

R. we are grateful for this comment. We have added the missing details of the study group, adding the mean age of both patients and controls, and the exclusion criteria. All patients were age- and sex-matched with the controls, thus reducing the bias effect of the age on the results of the neurophysiological test.

The discussion section should be shortened. The authors repeated the results without reference to the studies of other authors. They should discuss their points of view, and refer to the previous findings. We do not know whether the authors’ results are compatible or not with the previous studies. The discussion section must be rewritten.

The references cited should be enlarged.

R: we have rewritten the discussion, making it shorter and more readable; we have also added other missing and significant papers.

Reviewer 3 Report

Chronic inflammatory demyelinating polyneuropathy (CIDP) is a very heterogeneous disorder, with a wide spectrum of dysimmunological mechanisms, different clinical phenotypes, variable disease course and response to defined treatment. Thus, the search for new biomarkers is so useful in CIDP. In this study, the authors correlated clinical, conventional and unconventional neurophysiological parameters to identify sensitive biomarkers of axonal degeneration.

The article is correctly written, with a clear presentation of the study's aims, purposes and results. The subject is presented in detail in the discussion.  The results of the study may be useful in daily clinical practice. 

Author Response

We thank the reviewer for these positive comments to our work.

Reviewer 4 Report

The paper deals with neurophysiological hallmarks of axonal degeneration in CIDP patients assessed by unconventional ENG/EMG parameters. CIDP is a rather rare disease although it has been increasingly recognized in recent years due to the increasing availability of diagnostic methods.

This paper contains new knowledge in this area focuses on the search for new biomarkers of this disease. The biggest limitation of the paper is the very small study group and perhaps the title of the paper should include the term "pilot study"? I am not convinced that statistical methods performed on such a group are reliable. 

In addition, there has been recent discussion of different clinical phenotypes of CIDP and different responses to treatment. The authors should discuss this and refer to the following paper: https://pubmed.ncbi.nlm.nih.gov/35008604/

Author Response

The biggest limitation of the paper is the very small study group and perhaps the title of the paper should include the term "pilot study"? I am not convinced that statistical methods performed on such a group are reliable.

R: we thank the reviewer for these comments, we agree that this is a pilot study and we have changed the title, accordingly.

 In addition, there has been recent discussion of different clinical phenotypes of CIDP and different responses to treatment. The authors should discuss this and refer to the following paper: https://pubmed.ncbi.nlm.nih.gov/35008604/

R: we have rewritten the discussion, making it shorter and more readable; we have also added other missing and significant papers.